# Clinical Interpretability of Deep Learning for Predicting Microvascular Invasion in Hepatocellular Carcinoma by Using Attention Mechanism

**DOI:** 10.3390/bioengineering10080948

**Published:** 2023-08-09

**Authors:** Huayu You, Jifei Wang, Ruixia Ma, Yuying Chen, Lujie Li, Chenyu Song, Zhi Dong, Shiting Feng, Xiaoqi Zhou

**Affiliations:** Department of Radiology, The First Affiliated Hospital, Sun Yat-sen University, 58th the Second Zhongshan Road, Guangzhou 510080, China; youhy3@mail.sysu.edu.cn (H.Y.); wjif@mail.sysu.edu.cn (J.W.); marx3@mail2.sysu.edu.cn (R.M.); chenyy593@mail.sysu.edu.cn (Y.C.); lilujie@mail.sysu.edu.cn (L.L.); songchy5@mail2.sysu.edu.cn (C.S.); dongzh7@mail.sysu.edu.cn (Z.D.)

**Keywords:** hepatocellular carcinoma, microvascular invasion, attention mechanism, deep learning, multi-phase MRI

## Abstract

Preoperative prediction of microvascular invasion (MVI) is essential for management decision in hepatocellular carcinoma (HCC). Deep learning-based prediction models of MVI are numerous but lack clinical interpretation due to their “black-box” nature. Consequently, we aimed to use an attention-guided feature fusion network, including intra- and inter-attention modules, to solve this problem. This retrospective study recruited 210 HCC patients who underwent gadoxetate-enhanced MRI examination before surgery. The MRIs on pre-contrast, arterial, portal, and hepatobiliary phases (hepatobiliary phase: HBP) were used to develop single-phase and multi-phase models. Attention weights provided by attention modules were used to obtain visual explanations of predictive decisions. The four-phase fusion model achieved the highest area under the curve (AUC) of 0.92 (95% CI: 0.84–1.00), and the other models proposed AUCs of 0.75–0.91. Attention heatmaps of collaborative-attention layers revealed that tumor margins in all phases and peritumoral areas in the arterial phase and HBP were salient regions for MVI prediction. Heatmaps of weights in fully connected layers showed that the HBP contributed the most to MVI prediction. Our study firstly implemented self-attention and collaborative-attention to reveal the relationship between deep features and MVI, improving the clinical interpretation of prediction models. The clinical interpretability offers radiologists and clinicians more confidence to apply deep learning models in clinical practice, helping HCC patients formulate personalized therapies.

## 1. Introduction

Hepatocellular carcinoma (HCC) is the sixth most common cancer and the third leading cause of cancer-related mortality globally [1,2]. Worldwide, in 2019, there were approximately 747,000 cases of HCC, and 480,000 patients died due to HCC [3]. China has the greatest HCC burden, followed by Japan [3]. There is a wide variety of therapies for HCC, such as curative resection, radiofrequency ablation, and liver transplantation. However, the overall survival of HCC remains poor, with a medium survival of 6–10 months [3]. High recurrence and metastasis are the major causes of poor survival of patients with HCC [4].

Microvascular invasion (MVI) is a well-known prognostic factor for HCC and is positively associated with the HCC recurrence [5,6]. Reliable preoperative identification of MVI is beneficial for choosing optimal therapies, such as extended resection margins or a combination of postsurgical neoadjuvant therapy [7,8]. However, MVI is normally assessed using histopathological specimens after surgical resection [8,9]. Accurate preoperative prediction of MVI is of great significance in clinical decision making [9]. The motivation of this study is to develop a MVI prediction model with deep learning and offer it visual explanations, which can accelerate its application in the clinical practice and help HCC patients formulate personalized therapies, thus reducing the HCC recurrences and improving the overall survival.

Preoperative prediction of MVI remains a challenge [10]. Many studies produced MVI prediction models based on radiological features [5,6] and radiomic analysis [2,11]. Since imaging features rely on radiologists, interobserver variability would exist [10], and radiomics analysis is time-consuming as the tumor boundary needs to be manually defined [12]; therefore, the produced models lack practical feasibility. Multi-phase magnetic resonance imaging (MRI) can offer comprehensive anatomical and functional information about tumors, and the fusion of multi-phase MRI with deep learning in preoperative MVI prediction is trending [13,14]. Deep learning can automatically extract high-dimensional features and analyze the relationship between the image features and clinical problems. Many studies proposed MVI prediction models with deep learning based on multi-phase MRI and reported that multi-phase fusion models outperformed single-phase models [13,14]. Zhou et al. developed a multi-phase fusion model based on pre-contrast, arterial, and portal phases of contrast-enhanced MRI with a three-dimensional (3D) convolutional neural network (CNN) and found that the fusion model outperformed the single-phase models [13]. Similarly, a study by Zhang et al. fused multi-phase MRI with 3D CNN and demonstrated that a fusion model combining three sequences of MRI had better prediction performance than single-sequence models [14]. However, deep learning is an end-to-end process that directly shows the association between images and MVI but cannot explain the cause-and-effect relationship between deep features and outcomes. The black-box nature of deep learning makes it less interpretable, thereby obstructing its clinical application [15].

Attention mechanism was first proposed by Bahdanau et al. [16] for dealing with the difficulty that neural networks faced in processing long sentences. They developed a deep learning architecture called RNNsearch by implementing an attention module in the decoder, which allowed the network assign weights to different parts of the input data and select the most useful features, thus improving translation accuracy [16]. Features with larger weights are considered to be more valuable to outputs. Attention mechanism is widely used in the field of image processing and exhibits good performances, such as automatic facial expression recognition and tooth segmentation [17,18,19,20,21]. Recently, self-attention and collaborative-attention modules were integrated into deep learning to fuse features from multimodalities and help attention fusion models achieve better performance than other fusion models [22,23]. The self-attention module, also known as intra-attention block, is a structure that assigns attention weights to features based on their importance within the same modality and captures useful features [23]. The collaborative-attention module is a cross-modal inter-attention module and is used to explore the feature relations of different modalities after the self-attention module [23]. It combines related features from different modalities and highlights useful features, which helps deep learning networks focus on the most informative regions and select the most favorable features [21,23]. Furthermore, the weights provided by the attention modules are analytical and easily interpretable [21]. The visualization of attention weights can directly display salient regions to the final prediction, which offers insights into prediction models and improves its clinical interpretation [21,24]. This suggests that embedding attention mechanisms into deep learning may improve the prediction performance of MVI and reveal the underlying mechanism behind predictions, overcoming the black-box nature of deep learning. A study by Xu et al. adopted a self-attention-based vision transformer to predict MVI from CT images and their model produced an AUC of 0.88 [25]. Their attention maps generated from attention weights allow color visualization of the suspicious patches of MVI [25]. However, whether the implementation of self-attention and collaborative-attention in 3D CNN can effectively predict MVI based on multi-phase MRI and provide interpretation to deep learning has not been investigated.

Therefore, the purpose of this study was to investigate whether an attention-guided feature fusion network which includes intra- and inter-attention modules can accurately predict MVI based on multi-phase MRI. More importantly, we tried to reveal the cause-and-effect relationship between deep features and MVI by visualizing attention weights in the training network, thereby increasing the clinical interpretability of prediction models.

## 2. Material and Methods

In this section, we first introduce the recruitment process of the patients with HCC and the collection of clinical data. Second, the section presents the pathological criteria for MVI evaluation. Third, it introduces the aquation of MR images. Fourth, it describes the process of the models’ development, including the tumor segmentation in MR images, tumor cubic regions’ extraction, data augmentation and resizing, models’ training process, models’ evaluation, and models’ visualization. The flow diagram of the overall workflow is presented in Figure 1. 

### 2.1. Patients

This retrospective study was approved by the Ethical Review Authority, and the requirement for written informed consent was waived by the Ethical Review Authority due to the retrospective nature of this study.

From 28 September 2016 to 16 May 2022, 210 patients with primary solitary HCC (186 men and 24 women) who were enrolled at the First Affiliated Hospital, Sun Yat-Sen University, were recruited for this study. Patients were included if they met the following criteria: (a) underwent gadoxetate-enhanced MRI examination within 2 weeks before surgery, (b) underwent curative liver resection, and (c) were diagnosed as having HCC based on surgical specimens and pathological reports including MVI information. Patients were excluded for the following reasons: (a) multiple HCCs, (b) presence of macrovascular invasion (gross tumor thrombus), (c) underwent neoadjuvant therapy such as transarterial chemoembolization before surgical resection, (d) co-morbidity with other malignancies, (e) recurrence of HCC, and (f) suboptimal image quality for analysis.

The routine preoperative clinical characteristics of patients, including age, sex, alanine aminotransferase, aspartate aminotransferase, and alpha-fetoprotein, were obtained from clinical reports. Maximum tumor diameters were measured in the hepatobiliary phase (HBP) by two trained radiologists with 2 and 3 years of experience in MRI diagnosis.

According to the pathological reports, among the 210 patients 70 were MVI-positive (mean age, 54 years ± 13 [standard deviation]), and 140 were MVI-negative (mean age, 55 years ± 11 [standard deviation]). The patients’ demographics in MVI-positive and MVI-negative patients are shown in Table 1.

### 2.2. Pathological Records

The pathological sampling of HCC specimens was based on the baseline 7-point sampling scheme in the standardized pathological diagnosis guidelines for primary liver cancer [26]. HCC diagnosis was based on morphological criteria defined by the World Health Organization. MVI was defined as the presence of tumor emboli in a vascular space lined by endothelial cells in the peritumoral liver, which was visible by microscopy [1,10].

### 2.3. MRI Examination

MRI was performed using a 3.0-T MRI scanner (Magnetom Prisma; Siemens Medical Systems, Erlangen, Germany) with an eight-channel torso-array coil within 2 weeks before surgical resection. Before MRI examination, all patients fasted for 6–8 h. The liver MRI protocols are presented in Table 2. Dynamic contrast-enhanced MRI was performed after administering gadoxetate (0.1 mL/kg, Primovist, Bayer Schering Pharma, Berlin, Germany). The contrast agent was injected at a flow rate of 2 mL/s, and 30 mL of normal saline was flushed at the same rate. For dynamic sequences, arterial, portal, and hepatobiliary phases were obtained 25 s, 66 s, and 20 min after contrast media injection, respectively.

### 2.4. Deep Learning Models’ Development

The models’ development included several steps. First, we needed to segment tumors in MR images and extract 3D cubic tumors with bounding boxes. These 3D cubic tumors were separated into training and test sets randomly in 4:1 ratio. Extracted 3D cubic regions in the training set were resized and augmented to a large amount of small 3D cubic regions. Second, the 3D cubic regions in the training set were fed into the deep learning network to train models. The 3D cubic regions in the test set were used to validate the prediction performance. Third, the attention weights in the training networks were visualized to display the informative regions to the prediction decisions.

#### 2.4.1. Image Preprocessing and Data Augmentation

MVI was scattered in the 3D space of the peritumoral areas. The input included multilayer images for extracting the 3D features of tumors and peritumoral areas. To locate a tumor when extracting 3D tumor cubic regions with bounding boxes, firstly the tumors needed to be labeled. Tumor borders in the HBP were distinct and easy to recognize. Hence, we segmented the tumor in the HBP. Therefore, two trained radiologists with 2 and 3 years of experience in MRI diagnosis manually segmented the tumor borderlines at the top, largest, and bottom layers of the tumor on axial images of the HBP using ITK-SNAP software (www.itksnap.org) (Version 3.8.0-beta), (accessed on 28 October 2018) covering the entire tumor. A senior radiologist with 4 years of experience in MRI diagnosis reviewed all segmentations. The tumor borders in the arterial, portal, and pre-contrast phases were automatically matched based on the segmentation of the HBP.

After finishing tumor segmentations, a bounding box was centered on the volume of interest to cover the tumor and was expanded by 2 pixels to include peritumoral areas, extracting a cubic region. Then, 210 patients were randomly divided into training and test sets in a ratio of 4:1. Finally, 168 patients were randomly selected for training, and 42 patients were selected for testing. In the training set, image resampling included data resizing and augmentation, which was similar to a previous study [27]. This aimed to overcome the lack of large datasets and avoid the overfitting problem caused by insufficient data in the training set, so the images data in the training set underwent augmentation to obtain local region-based features of MR images. In the data-resizing step, the original cubic region was resized to the same size (16 × 16 × 16) to acquire features based on the global regions of MR images. A total of 168 global cubic regions were obtained in the training set. In the data augmentation step, the original cubic region was first resized into a predetermined size (32 × 32 × 32) and then cut into a small size (16 × 16 × 16) in the x, y, and z directions with an interval of 2 pixels. Further, 9, 9, and 9 cubic regions along the x, y, and z directions, respectively, were generated in one tumor cubic region. Finally, 729 local cubic regions were generated from each tumor in a single phase. Overall, 122,640 samples for each phase, including 729 × 168 local cubic regions and 168 global cubic regions, were generated for model training. The data in the test set were resized to the same size (16 × 16 × 16), and 42 global region samples in the test set were used for model evaluation. For the test set, data augmentation would make a large amount of the augmented data share the same MVI label, which would introduce data bias. So, the samples in the test set only underwent data resizing. A flowchart of the image preprocessing and data augmentation is shown in Figure 2.

#### 2.4.2. Model Development

This study was motivated by a previous study by Li et al. [23]. They proposed an attention-guided discriminative feature learning and fusion framework which included a cross-modal intra- and inter-attention module for grading HCC and demonstrated that the proposed method outperformed the previous feature fusion methods [23]. The cross-modal intra- and inter-attention module considers the intra- and inter-relation among modalities to harness the complementary information among different phases, which can help deep learning focus on the most informative region of each modality and select the most useful features for predictions [23]. Moreover, attention modules can provide attention weights that can be visualized and display salient regions to predictions, increasing the clinical interpretability of prediction models [21,24]. Therefore, the motivation of this study was to apply the attention-guided feature fusion network proposed by Li’s group [23] to develop MVI prediction models based on multi-phase MRI and to acquire visual explanations of predictions through the visualization of weights offered by attention modules. 

The framework of the deep learning network used for single-phase models is shown in Figure 3A and was implemented in the pre-contrast (PreP), arterial (AP), portal (PP), and HBP model. Attention-guided feature fusion network, as shown in Figure 3B, was implemented in the arterial + portal phase model (2P), the arterial + portal + hepatobiliary phase model (3P), and the pre-contrast + arterial + portal + hepatobiliary phase model (4P). In this framework, self-attention and collaborative-attention blocks were added between the convolutional and pooling layers on the deep learning network LeNet5. The frameworks of the self-attention and collaborative-attention blocks are shown in Figure 4 and Figure 5, respectively. The purpose of the self-attention block is to assign high-attention weights to MVI-relevant features and low-attention weights to MVI-irrelevant features. The collaborative-attention block complements related information from other phases and adjusts attention weights to further highlight the relevant features.

The 3D cubic regions were fed into the network. The extracted features were concatenated and flowed into fully connected layers and a SoftMax layer to generate the final results. The network training was repeated for 5 runs due to the 5-fold cross-validation schemes. The batch size of each network was 16, and each model was trained for 50 epochs using a momentum-based stochastic gradient descent optimizer (learning rate starting: 0.004, momentum parameter: 0.9, weight decay: 0.0001). The maximum number of iterations was set to 501.

The loss function in this study is L_classifify_ = ∑ω_x_L_x_. For single-phase models, there is only one ω, while for multi-phase models, there are 2 ω for two-phase models, 3 ω for three-phase models, and 4 ω for four-phase models. The fusion loss function measures the discrepancy between the model’s predicted outputs and the ground truth annotations. During the training process, the attention mechanism is adjusted by backpropagating the gradients through the model to update the weights ω and improve the alignment between the input and output sequences. 

Furthermore, we employed the following techniques to prevent overfitting:Data augmentation: We augmented the 168 original training samples to 122,640, allowing the model to have sufficient numbers of validation data to help the model converge in the 5-fold cross-validation session.Dropout: Dropout randomly removes a fraction of the neurons during the training process, forcing the model to be less reliant on specific neurons and preventing potential overfitting.Early stopping: We employed early stopping during the model training process. By outputting the learning curve in real time during the training process, we could monitor the model’s performance on a validation set and stop the training process once the performance started to deteriorate, which can prevent the model from excessively fitting to the training data.

All models were developed with Python Software Foundation (28 June 2021). Python Language Reference (version 3.9.6) https://www.python.org/downloads/release/python-396/, PyTorch (Version 1.12.1), (accessed on 28 June 2021). PyTorch: Tensors and dynamic neural networks in Python with strong GPU acceleration. https://pytorch.org/docs/1.12/, (accessed on 28 June 2022).

#### 2.4.3. Model Evaluation and Features Visualization

Receiver operating characteristic (ROC) curves were used to evaluate the predictive performance of each model. The area under the curve (AUC) and 95% confidence interval (CI), accuracy, sensitivity, and specificity were recorded. Decision curve analysis (DCA) was performed to compare the clinical benefits of different models. To understand the underlying prediction mechanism, we generated activation heatmaps with gradient-weighted class activation mapping (Grad-CAM) based on the attention weights in the self-attention and collaborative-attention layers [28]. After concatenation, the tensor weights in the fully connected layers of each phase were visualized using heatmaps to demonstrate the contributions of each phase to the multi-phase fusion models.

### 2.5. Statistical Analysis

Statistical analysis was performed using the Statistical Package for the Social Sciences software (version 25.0; IBM SPSS, Inc., Chicago, IL, USA). The Shapiro–Wilk test was used to test the normality of continuous variables. Continuous variables were analyzed using Student’s t-test or the Mann–Whitney U test based on their normality and expressed as mean ± standard deviation or median (interquartile range). Categorical variables were analyzed using Pearson’s chi-squared test and expressed as numbers (percentages). A two-tailed *p* value < 0.05 was considered statistically significant.

## 3. Results

### 3.1. MVI Prediction Performance of the Deep Learning Models

Two hundred and ten patients were randomly separated into the training (*n* = 168) and test (*n* = 42) sets. In the training set, there were 56 MVI-positive and 112 MVI-negative patients. In the test set, there were 14 MVI-positive and 28 MVI-negative patients. The characteristics of patients in the training and test sets are showed in Table 3.

The AUC, accuracies, sensitivities, and specificities of the different models in the test set are shown in Table 4. The PreP, AP, PP, and HBP models had AUCs of 0.75 (95%CI: 0.59–0.90, *p* = 0.010), 0.82 (95%CI: 0.68–0.96, *p* = 0.001), 0.82 (95%CI: 0.69–0.95, *p* = 0.001), and 0.86 (95%CI: 0.73–0.98, *p* < 0.001), respectively. The 2P model had an AUC of 0.84 (95%CI: 0.73–0.96, *p* < 0.001), and the 3P model achieved an AUC of 0.91 (95%CI: 0.81–1.00, *p* < 0.001). The 4P model had the best prediction performance (AUC, 0.92; 95%CI: 0.84–1.00, *p* < 0.001).

The ROC and DCA curves of the different models in the test set are shown in Figure 6. The ROC curves showed that the 4P model had the best predictive performance (Figure 6A). The DCA curves showed that the 3P and 4P models had more clinical benefits than the 2P and single-phase models (Figure 6B). In a large range of threshold probabilities, the 4P model can always add more clinical benefits than the single-phase models.

### 3.2. Visualization and Interpretation of the Deep Learning Models

To improve the clinical interpretation of deep learning, we generated activation heatmaps using Grad-CAM based on attention weights in the self-attention and collaborative-attention layers. The representative example is presented in Figure 7. Figure 7A–D are the original MR images of the pre-contrast, arterial, portal, and hepatobiliary phases.

The heatmaps of the self-attention layers showed that the tumor and peritumoral areas in the arterial and hepatobiliary phases gained high attention, whereas in the portal phase only the tumor gained high attention (Figure 7F–H). The pre-contrast phase did not show a specific area to focus on (Figure 7E). After fusing the arterial and portal phases, the heatmaps of the collaborative-attention layers showed that the border of the tumor in the arterial phase and the tumor body in the portal phase had high-attention weights (Figure 7I,J). In the 3P model, the heatmaps of the collaborative-attention layers showed that the tumor margins and peritumoral areas in the three phases received high attention (Figure 7K–M). After fusing relevant information from the four phases, the heatmaps of the arterial and hepatobiliary phases showed that the tumor margins and peritumoral areas acquired high attention (Figure 7O,Q), while the heatmaps of the pre-contrast and portal phases showed that the tumor margins gained high attention (Figure 7N,P).

To demonstrate the relative importance of the extracted features in each phase, the tensor weights of each phase were visualized using heatmaps after direct concatenation, as shown in Figure 8. The results showed that in the 2P model the weights in the arterial phase were higher than those in the portal phase, whereas in the 3P and 4P models the weights in the HBP were much higher than those in the other three phases.

## 4. Discussion

In this study, we applied an attention-guided feature fusion network to develop MVI prediction models based on multi-phase MRI and found that the 4P model had the best prediction performance (AUC, 0.92; 95%CI: 0.84–1.00). The visualization of attention weights provided by the attention modules revealed that the tumor margins in the four phases and peritumoral areas in the arterial and hepatobiliary phases were the most relevant regions contributing to MVI prediction. Simultaneously, heatmaps of tensor weights in the fully connected layers indicated that the HBP was the most important phase contributing to MVI prediction.

In this study, the 4P model achieved an AUC of 0.92, which is comparable to the excellent deep learning models reported in previous studies that produced an AUC of 0.91 to 0.92 without using attention mechanism [13,15,29]. The common shortcoming of these studies is that they cannot give interpretability to their prediction models. Recently, a study by Xu et al. proposed a MVI prediction model with a vision transformer-based pure self-attention structure using contrast-enhanced CT images and demonstrated that the model achieved an AUC of 0.88 [25]. Similarly, Li et al. developed a modality-based attention and dual-stream multiple instance CNN for predicting MVI and reported that the model produced an AUC of 0.74 [30]. Both these studies used the self-attention module to develop MVI prediction models, but their models achieved a relatively low performance. This may be attributable to the fact that they did not combine the collaborative-attention module after the self-attention module, indicating that the collaborative-attention module plays a crucial role in improving MVI prediction performance. In addition to our study, several studies also demonstrated that self-attention and collaborative-attention modules can enhance prediction performance during imaging processing in other different fields, such as skin lesion diagnosis and grading HCC [22,23]. The underlying mechanism is that during the training of deep learning, self-attention blocks assign attention weights to different features within the same phase, and collaborative-attention blocks learn related features from other phases and assist the deep learning network in paying attention to key features by suppressing attention weights to irrelevant features and adding attention weights to relevant features [22]. The highlight of the attention-guided feature fusion network is that the collaborative-attention block considers the intrinsic association between different phases, which is consistent with the clinical logic that images from different phases are not observed separately in the evaluation of MVI. The arterial, portal, and hepatobiliary phases reflect different perfusion statuses and can offer different types of information on the tumor and peripheral liver parenchyma [31]. Hence, the multi-phase fusion method is important for improving performance in MVI prediction. The attention-guided feature fusion model achieved a promising performance in MVI prediction, suggesting that this network is suitable for developing MVI prediction models based on multi-phase MRI.

The black-box nature of deep learning leads to a poor understanding of the relationship between deep features and outcomes. To understand the association between deep features and MVI, we generated attention heatmaps using Grad-CAM based on weights in the self-attention and collaborative-attention layers. The attention heatmaps of the collaborative-attention layers in the 4P model showed that the tumor margins in the four phases and the peritumoral areas in the arterial and hepatobiliary phases had high-attention weights, indicating that these areas contributed the most to MVI prediction. Recently, a study by Xu et.al. [25] produced a MVI prediction model with the self-attention module based on CT images and generated heatmaps with attention weights. Their studies revealed that attention maps can provide color visualization in detecting suspicious MVI patches [25]. Similarly, Xiao et al.’s [32] group proposed an expert-inspired MVI prediction model based on CT images and generated an activation map based on the final model. The activation map in their study showed that discontinuous borderline and peritumoral enhancement were strong predictive factors for MVI [32]. Zeng et al. developed a MVI prediction model with an attention mechanism based on intravoxel incoherent motion diffusion-weighted imaging, and the model achieved an AUC of 0.85 [33]. The attention map of their study showed that main parts of the tumor received high attention, in contrast with the results of our study. The main reason may be that the weak borders and noisy backgrounds of diffusion-weighted imaging reduced the significance of the tumor margins [33]. This study revealed an intuitive understanding of the association between deep features and MVI, which increased the clinical interpretability of prediction models and addressed the black-box nature of deep learning.

In addition, the high-attention regions contributing to MVI prediction in the four phases are consistent with the radiological features reported in previous studies [10,34]. Many previous studies demonstrated that radiological features such as nonsmooth tumor margins, rim arterial enhancement, peritumoral hyperenhancement in the arterial phase, and peritumoral hypointensity in the HBP were significant features for MVI prediction [5,6,9,10,35]. According to a study by Jiang’s group, tumor margins ranked first among 129 radiomics–radiological–clinical features, indicating that tumor margins were the most important radiological feature in identifying MVI [29]. Pathologically, irregular tumor margins indicate nodules protruding into the non-tumor parenchyma, which represents a more aggressive tumor growth pattern and increases the occurrence of MVI [9,36]. Irregular rim-like hyperenhancement in the arterial phase is associated with more aggressive pathological features [36,37]. MVI mainly occurs in the minute portal vein branch, leading to hemodynamic perfusion changes in the peritumoral areas, in which portal flow decreases and arterial flow compensatively increases [5,38,39]. Hemodynamic perfusion changes further cause the functional impairment of organic anion-transporting polypeptide transporters in peritumoral hepatocytes [6,36], which induces peritumoral hypointensity in the HBP. The MVI-associated imaging features and pathological characteristics provided radiological and pathological explanations for our findings and revealed a cause-and-effect relationship between deep features and MVI, which further increased the clinical interpretability of our models.

In the present study, heatmaps of tensor weights in the fully connected layers after concatenation showed that in the 3P and 4P models, the HBP showed the highest tensor weights, indicating that the HBP was the most important phase contributing to MVI prediction. This finding was confirmed by the fact that the HBP model exhibited the best performance among four single-phase models. In addition, heatmaps of tensor weights showed that the contribution of the pre-contrast and portal phases to the final prediction was much smaller than that of the arterial and hepatobiliary phases, which was in line with result that the PreP and PP models had lower predictive performances than the AP and HBP models. After fusing the relevant features from other phases, the pre-contrast and portal phases learned to pay more attention to tumor margins. Visualization of the tensor weights in the fully connected layers provided information about the contributions of each phase to the final prediction, providing clinical interpretability of deep learning models. This study suggested that among the four phases, the HBP is most important in evaluating MVI.

Our prediction model exhibits a promising performance in MVI prediction, which offers a non-invasive method to preoperatively predict MVI. By using our model, radiologists and clinicians can preoperatively obtain the evidence of MVI in patients with HCC. The color visualization of the model highlights the areas that contain critical information, enabling radiologists and clinicians to focus on these areas and provides supplemental information for MVI prediction. The high-attention regions in the MR images may also show the potential usefulness in guiding pathologists in sampling. Moreover, our prediction model can be explained by radiological features, allowing radiologists and clinicians to confidently apply it in clinical settings. Finally, our study found that the HBP is the most important phase in evaluating MVI, suggesting that radiologists and clinicians should pay more attention to the HBP, which may be helpful in finding evidence of MVI existence. The combination of deep learning and radiological features can increase the accuracy of MVI prediction and save radiologists’ time in evaluating MVI. Preoperative identification of MVI is crucial for patients with HCC to formulate personalized therapies, which can reduce the HCC recurrence and improve the overall survival [30].

Our study has several limitations. First, this is a retrospective study, which might induce a potential patient selection bias. A large prospective clinical trial is necessary to validate deep learning models in clinical practice. Second, this study was conducted at a single healthcare center in China without an external test set, limiting the model’s generalizability. A larger number of study groups from multiple centers involving varying types of HCC or MVI in different ethnic groups or populations is needed to validate our prediction models. Third, patients with HCC included in this study had solitary tumors. Multiple tumors are frequently associated with a higher incidence of MVI and are much more complicated; therefore, our prediction model cannot directly apply to patients with multiple HCCs. MVI prediction models based on multiple HCCs require further investigation.

## 5. Conclusions

The attention-guided multi-phase fusion network performed excellently in preoperative MVI prediction. Tumor margins in the four phases and the peritumoral areas in the arterial and hepatobiliary phases were the most salient regions contributing to MVI prediction. The HBP contributed the most to MVI prediction among the four phases. Visualizing the attention weights in the training network provides an intuitive understanding of the cause-and-effect relationship between deep features and MVI, increasing the clinical interpretation of deep learning models, which may facilitate the clinical application of deep learning. Accurate preoperative prediction of MVI can help screen patients with HCC who are at risk of MVI, assisting clinicians in selecting more suitable therapies to achieve better survival.

## Figures and Tables

**Figure 1 bioengineering-10-00948-f001:**
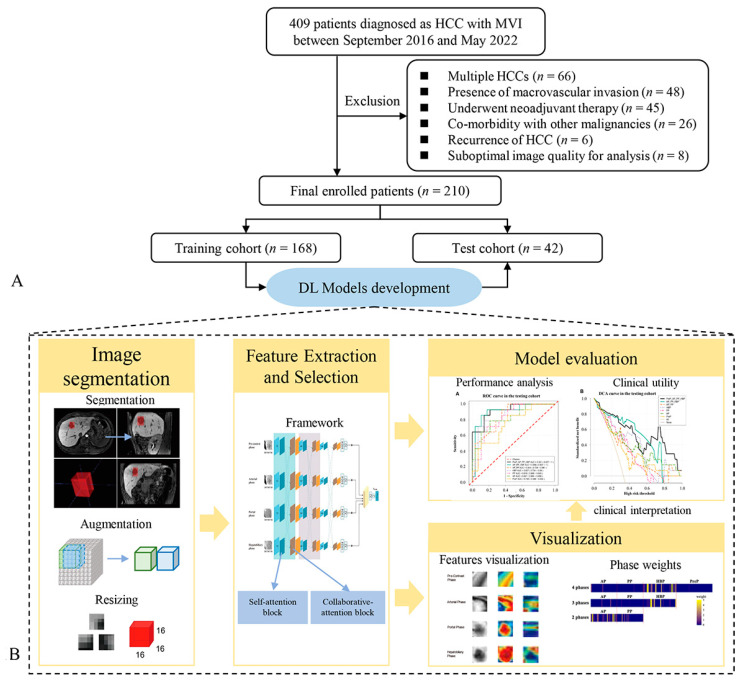
Flow diagram depicting patients’ recruitment and the deep learning workflow. (**A**) describes the process of patients’ recruitment, and (**B**) demonstrates the flowchart of models’ development and visual explanation, including tumor segmentation, tumor cubic regions extracting, models’ training and validation, model evaluation, visualization of attention weights. HCC, hepatocellular carcinoma; MVI, microvascular invasion.

**Figure 2 bioengineering-10-00948-f002:**
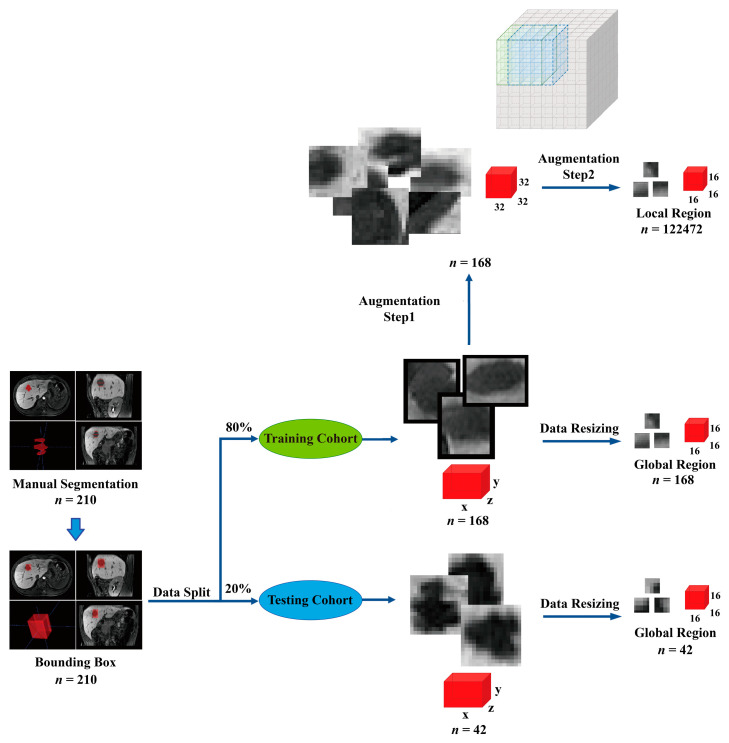
Image preprocessing and data augmentation. The tumor was segmented at the top, largest, and bottom layers of tumor on axial images of the hepatobiliary phase covering the entire tumor. A bounding box (red) was then centered on the volume of interest to extract a cubic region containing the tumor and was expanded by 2 pixels to include peritumoral areas. The patients were randomly divided into training (*n* = 168) and test sets (*n* = 42) in a ratio of 4:1. The data in the training set were resized into the same size (16 × 16 × 16) for extracting global features. Furthermore, the data in the training set were also resized into a predetermined size (32 × 32 × 32) and cut into a small size (16 × 16 × 16) in the x, y, and z directions with an interval of 2 pixels for extracting local features. The gray box is the schematic diagram indicating the process of data augmentation. The data in the test set were resized into the same size (16 × 16 × 16) for model evaluation. The red colors box represents tumor cubic regions in different sizes.

**Figure 3 bioengineering-10-00948-f003:**
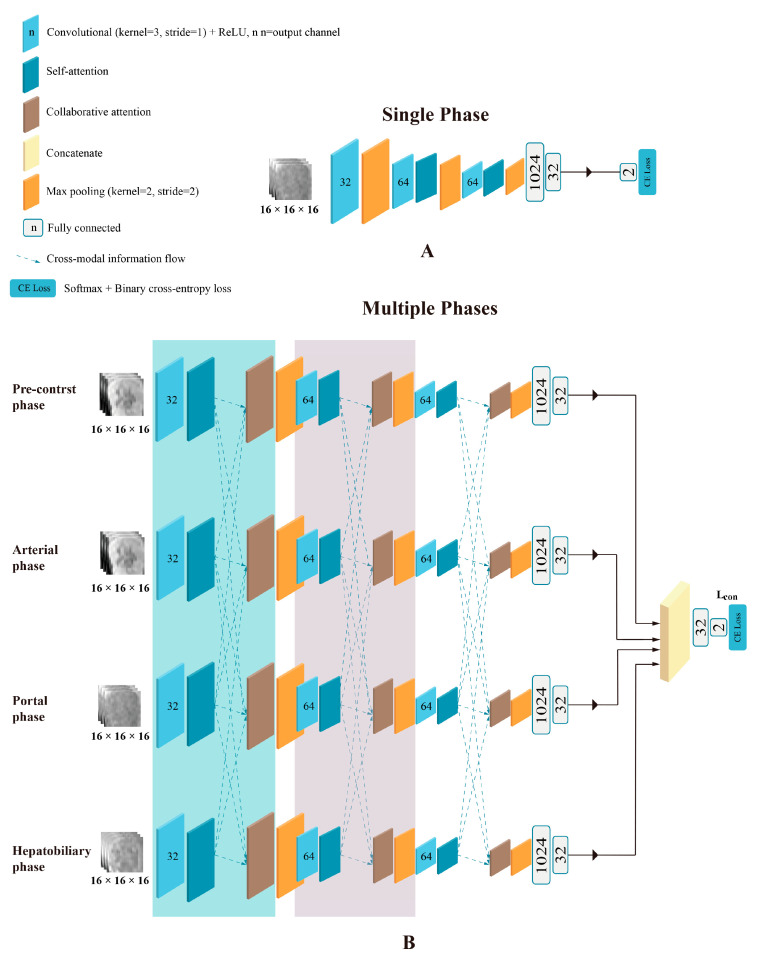
Framework of the deep learning network used in this study. (**A**) Framework used in single-phase models was implemented in the pre-contrast, arterial, portal, and hepatobiliary phase models. (**B**) Framework of attention-guided feature fusion network used for the pre-contrast + arterial + portal + hepatobiliary phase model (4P). The arterial + portal phase model (2P) and the arterial + portal + hepatobiliary phase model (3P) used similar frameworks. The self-attention block and collaborative-attention block were added between the convolutional and pooling layers.

**Figure 4 bioengineering-10-00948-f004:**
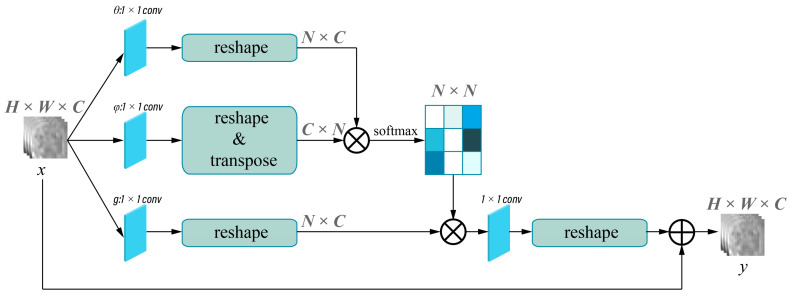
Self-attention block. X indicates the input images, and y indicates the output images of the self-attention layer. Input size of the image is H × W × C. H, W, and C represent the height, the width, and the output channel of the image, respectively. N = H × W. The different shade colors in N × N indicates the obtained attention coefficient (0–1), and dark colors indicates higher attention coefficient.

**Figure 5 bioengineering-10-00948-f005:**
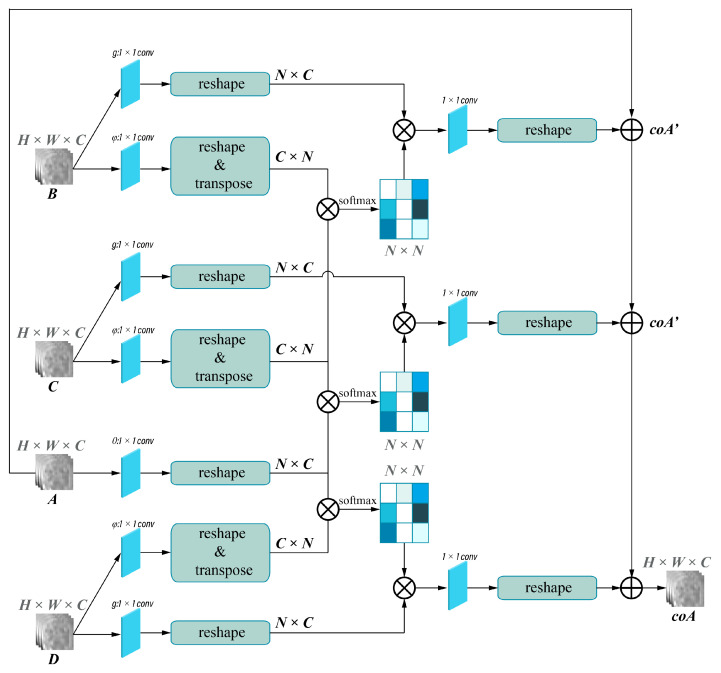
Collaborative-attention block. A, B, C, and D indicates input images of different phases from self-attention layers, and coA indicates output images of collaborative-attention layers. Input size of the image is H × W × C. H, W, and C represent the height, the width, and the output channel of the image, respectively. N = H × W. The different shade colors in N × N indicates the obtained attention coefficient (0–1), and dark colors indicates higher attention coefficient.

**Figure 6 bioengineering-10-00948-f006:**
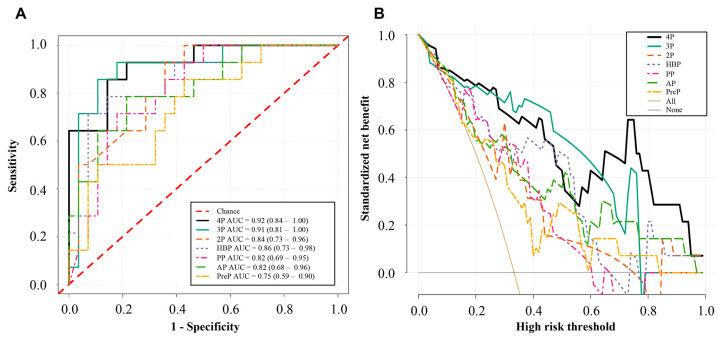
Receiver operating characteristic curves (**A**) and decision curve analysis (**B**) of different models in the test set. PreP represents the pre-contrast phase model; AP represents the arterial phase model; PP represents the portal phase model; HBP represents the hepatobiliary phase model; 2P represents the fusion model of the arterial and portal phases; 3P represents the fusion model of the arterial, portal, and hepatobiliary phases; 4P represents the fusion model of the four phases.

**Figure 7 bioengineering-10-00948-f007:**
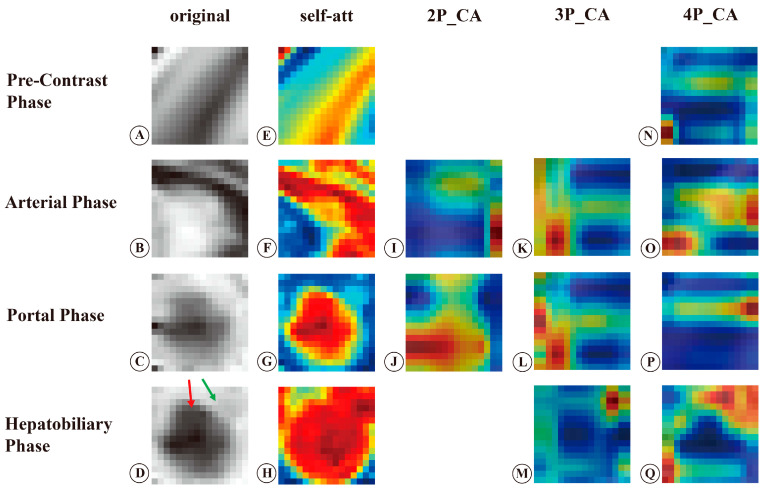
Original MR images and activation heatmaps in a 44-year-old woman with HCC and MVI (+). (**A**–**D**) are original MR images of the tumor in the pre-contrast, arterial, portal, and hepatobiliary phases. (**E**–**H**) are activation heatmaps of the self-attention layers in different phases. (**I**,**J**) are activation heatmaps of collaborative-attention layers in the 2 phase models. (**K**–**M**) are the activation heatmaps of collaborative-attention layers in the 3 phase models. (**N**–**Q**) are activation heatmaps of collaborative-attention layers that fused relevant information from all phases. The red arrow in (**D**) indicates the tumor, and the green arrow indicates peritumoral areas. The red color indicates high-attention weights, and the blue color indicates low-attention weights. Self-att, self-attention; CA, collaborative-attention; 2P, fusion model of the arterial and portal phases; 3P, fusion model of the arterial, portal, and hepatobiliary phases; 4P, fusion model of the pre-contrast, arterial, portal, and hepatobiliary phases.

**Figure 8 bioengineering-10-00948-f008:**
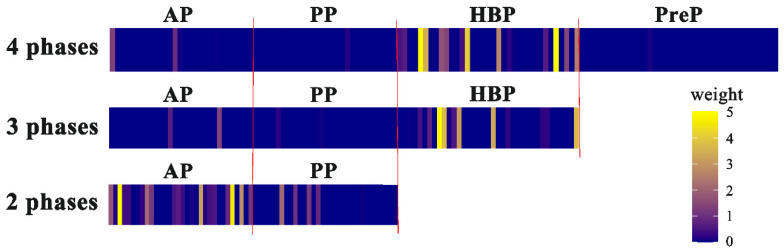
Heatmaps of tensor weights in three multi-phase fusion models after concatenation. Each phase had 32 grids. Bright color indicates high-tensor weights, and dark color indicates low-tensor weights. AP, arterial phase; PP, portal phase; HBP, hepatobiliary phase; PreP, pre-contrast phase.

**Table 1 bioengineering-10-00948-t001:** Clinical characteristics in MVI-positive and MVI-negative patients with HCC.

Parameters	MVI-Negative (*n* = 140)	MVI-Positive (*n* = 70)	*p*-Value
Age (years) *	55 ± 11	54 ± 13	0.818
Sex			>0.99
Men	124 (88.6)	62 (88.6)	
Women	16 (11.4)	8 (11.4)	
Aspartate aminotransferase (U/L) ^†^	32.0 (24.2–48.0)	33.0 (24.0–60.0)	0.378
Alanine aminotransferase (U/L) ^†^	33.5 (23.0–53.7)	32.0 (20.7–51.2)	0.682
Alpha-fetoprotein			<0.001
≥400 µg/L	28 (20.0)	30 (42.9)	
<400 µg/L	112 (80.0)	40 (57.1)	
Tumor diameter (mm) ^†^	42.5 (30.0–65.0)	57.0 (41.5–91.2)	<0.001

* Datum is means ± standard deviation. ^†^ Data are medians and numbers in parentheses are interquartile range. Sex and alpha-fetoprotein were compared with Pearson’s chi-squared test and expressed as numbers (percentages).

**Table 2 bioengineering-10-00948-t002:** MRI sequences in this study.

Sequences	TR (ms)	TE (ms)	FA	Thickness (mm)	Matrix	FOV (mm)	Breath-Hold
T1-weighted imaging	2.75	1.05	12.5	2	320 × 192	380 × 380	Yes
T2-weighted imaging	2000	77	103	5	384 × 288	380 × 380	No
Arterial/Portal phase	2.75	1.05	12.5	2	320 × 192	380 × 380	Yes
Hepatobiliary phase	3.84	1.45	25	2	288 × 186	380 × 306	Yes

Abbreviations: TR, repetition time; TE, echo time; FA, flip angle; FOV, field of view.

**Table 3 bioengineering-10-00948-t003:** Clinical characteristics in training and test sets.

Parameters	Training Set (*n* = 168)	Test Set (*n* = 42)	*p*-Value
Age (years) *	54 ± 11	56 ± 12	0.266
Sex			0.914
Men	149	37	
Women	19	5	
Aspartate aminotransferase (U/L) ^†^	33.0 (24.0–50.8)	31.0 (24.0–62.0)	0.933
Alanine aminotransferase (U/L) ^†^	33.0 (21.0–49.8)	32.0 (23.0–77.0)	0.469
Alpha-fetoprotein			0.354
≥400 µg/L	124	28	
<400 µg/L	44	14	
Tumor diameter (mm) ^†^	47.5 (34.0–70.0)	52.1 (31.8–71.0)	0.701

* Datum is means ± standard deviation. ^†^ Data are medians and numbers in parentheses are interquartile range. Sex and alpha-fetoprotein were compared with Pearson’s chi-squared test and expressed as numbers (percentages).

**Table 4 bioengineering-10-00948-t004:** Prediction performance of different models in the test set.

Model	AUC (95%CI)	Accuracy	Sensitivity	Specificity	*p*-Value
PreP	0.75 (0.59–0.90)	0.64	0.86	0.57	0.010
AP	0.82 (0.68–0.96)	0.74	0.79	0.79	0.001
PP	0.82 (0.69–0.95)	0.79	0.71	0.82	0.001
HBP	0.86 (0.73–0.98)	0.71	0.79	0.86	<0.001
2P	0.84 (0.73–0.96)	0.81	0.93	0.64	<0.001
3P	0.91 (0.81–1.00)	0.88	0.93	0.82	<0.001
4P	0.92 (0.84–1.00)	0.81	0.93	0.79	<0.001

Abbreviations: AUC, the area under the curve; CI, confidence interval.

## Data Availability

The code used for the modeling and data analysis is available at https://github.com/Grace4Wang/multiPhase_att (accessed on 29 March 2023). The datasets analyzed during the current study are available from the corresponding author on reasonable request.

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
