# Peer review of "Clinical Interpretability of Deep Learning for Predicting Microvascular Invasion in Hepatocellular Carcinoma by Using Attention Mechanism"

_bioengineering, 2023, doi:10.3390/bioengineering10080948_

Round 1
Reviewer 1 Report
The authors present a well written study on using self- and collaborative attention mechanisms on a deep learning model to detect microvascular invasion in liver cancer.
This proposed modelling method allows for better explainability.
It would have been an even stronger paper if the authors provided the code they used.
I believe that mentioning the programming language (Python?) and the deep learning framework (PyTorch?) they used is important and needs to be added.
I would also like to see a better explanation of what attention is, also what self attention and what collaborative attention is. This paper might be read by people who are less familiar with these terms, so adding a couple of sentences explaining what these terms are would be useful.
Reviewer 2 Report
Summary: The study utilizes an attention-guided feature fusion network to enhance the prediction of microvascular invasion (MVI) in hepatocellular carcinoma (HCC) patients preoperatively. Involving 210 HCC patients who received gadoxetate-enhanced MRI examination before curative resection, the retrospective study reveals improved predictive performance with multi-phase MRI over single-phase analyses. Remarkably, a four-phase fusion model demonstrated the highest efficiency.
Refined Detailed Comments:
1. I encourage a broader introduction that includes data about the prevalence and mortality rate associated with HCC to establish context and convey the importance of the study.
2. Expand the literature review in the introduction to cover additional relevant studies on MVI detection to highlight the novelty and necessity of the proposed approach.
3. Consider relocating "Patients’ demographics" to the beginning of the "Material and methods" section for a smoother narrative flow.
4. Lines 135-137: Clarify the process of deriving the number of cubic regions used for training and testing datasets. The reasoning behind selecting 42 samples for model evaluation requires additional elaboration.
5. Figure 2: Enhance clarity by providing a more detailed, larger visualization of the process for a specific tumor, inclusive of data augmentation and resizing.
6. Attention mechanisms: Explain how the loss function assesses and adjusts the attention mechanism within the model architecture. Further, define which features in the manual annotation are used for evaluating the performance of the attention mechanism.
7. It would be insightful if you could elaborate on the methods utilized to ensure that the model is not overfitting.
8. While the authors present a detailed account of the attention-guided feature fusion network's performance, a more precise outline of the model's overall architecture would further enhance understanding.
9. Emphasize the practical implications of your findings by discussing how clinicians could incorporate these models into their decision-making processes and the potential impacts on patient outcomes.
10. While the study results align with previous findings and provide additional insights through the use of the attention-guided feature fusion network, a more comprehensive comparison with existing literature would further validate your work.
11. The limitations discussion is thorough but does not address the model's generalizability to diverse ethnic groups or populations with varying HCC or MVI prevalence. Consider including this aspect for a more complete analysis of potential limitations.
Reviewer 3 Report
- The title is a bit ambiguous. What exactly was researched here? The authors propose a new attention mechanism or they simply perform a literature review? In 2023, the attention mechanism is already an old tool. Please rephrase.
- Please update the abstract section and highlight the novel contributions proposed in this manuscript. The formulation could also be improved. In my opinion, the current version is closer to a conclusion section.
- The manuscript’s motivation should be better introduced in the first paragraph of Section 1.
- The literature review is a bit short and “glued” to the quite short introduction.
- Section 2, please add a paragraph commenting on the connections between all its subsections.
- It’s not clear why Figure 1 was introduced. It could have been much more easily introduced as a table where the first column contains the numbers and the second column the added comments. It’s also a bit too large. The captions should also be updated to provide a bit more detail.
- Section 2.4., please add some comments on the general ideas studied in this manuscript.
- Section 2.4.1., please add an explanation on why such techniques are needed.
- Section 2.4.2., please add also some motivation for proposing such models. Whys this architecture provides good results? What is the idea behind its development?
- Comparison with state-of-the-art methods? Is it possible to have some comments on how similar are these results compared with other articles published in the same research domain?
- Please extract a conclusion section from the last part of the Discussion section.
The manuscript contains some typos, but one can follow the ideas.
Round 2
Reviewer 3 Report
The authors adequately answered all my questions.
The manuscript must be further checked.